# Infrared-Visible Image Fusion Based on Semantic Guidance and Visual Perception

**DOI:** 10.3390/e24101327

**Published:** 2022-09-21

**Authors:** Xiaoyu Chen, Zhijie Teng, Yingqi Liu, Jun Lu, Lianfa Bai, Jing Han

**Affiliations:** Jiangsu Key Laboratory of Spectral Imaging and Intelligent Sense, Nanjing University of Science and Technology, Nanjing 210094, China

**Keywords:** infrared-visible image fusion, semantic guidance, visual perception, intelligent vehicles

## Abstract

Infrared-visible fusion has great potential in night-vision enhancement for intelligent vehicles. The fusion performance depends on fusion rules that balance target saliency and visual perception. However, most existing methods do not have explicit and effective rules, which leads to the poor contrast and saliency of the target. In this paper, we propose the SGVPGAN, an adversarial framework for high-quality infrared-visible image fusion, which consists of an infrared-visible image fusion network based on Adversarial Semantic Guidance (ASG) and Adversarial Visual Perception (AVP) modules. Specifically, the ASG module transfers the semantics of the target and background to the fusion process for target highlighting. The AVP module analyzes the visual features from the global structure and local details of the visible and fusion images and then guides the fusion network to adaptively generate a weight map of signal completion so that the resulting fusion images possess a natural and visible appearance. We construct a joint distribution function between the fusion images and the corresponding semantics and use the discriminator to improve the fusion performance in terms of natural appearance and target saliency. Experimental results demonstrate that our proposed ASG and AVP modules can effectively guide the image-fusion process by selectively preserving the details in visible images and the salient information of targets in infrared images. The SGVPGAN exhibits significant improvements over other fusion methods.

## 1. Introduction

Driving assistance at night has always been a focus for intelligent vehicles. With the development of sensor technology, long-wave infrared cameras have been applied in intelligent systems to make up for the poor night vision performance of visible-light cameras. Due to the different characteristics of the sensors, there exist significant differences between visible-light images and infrared images. Visible images usually possess rich textures with a higher resolution compared with infrared images. However, the image quality can be easily affected by the surrounding conditions. In contrast, infrared images are captured based on the temperature characteristics or emissivity of the object. Targets with significant thermal radiation characteristics in infrared images are more salient, which makes it much easier for humans to find the targets. Although infrared images provide relatively fewer details, the image quality is not in-line with the perception of human eyes. Therefore, combining two complementary images to obtain a fused image with high quality is strongly demanded by many practical applications.

Over the past decades, many image fusion methods have been proposed [1]. These methods can be roughly divided into two categories: traditional image-fusion algorithms and deep-learning-based image-fusion algorithms [2]. In particular, traditional methods consist of multi-scale transform image fusion [3], sparse representation image fusion [4], low-rank representation image fusion [5], subspace-based image fusion [6], and saliency-based image fusion [7]. However, traditional algorithms are mainly hand-crafted. To achieve a better fusion result, the traditional algorithms are becoming more and more complex, which leads to the difficulty of practical applications and tedious computing time. In addition, a large number of traditional algorithms ignore the semantic information of the image and the saliency of the targets, resulting in fuzzy and illegible targets in fusion images.

Therefore, with the rise of deep learning and neural networks over recent years, deep-learning techniques applied to infrared and visible image-fusion methods have also emerged [8]. Deep-learning-based methods can be broadly classified into two categories: neural-network-based image fusion with end-to-end training and generative adversarial-network-based image fusion. Generative adversarial networks (GAN) are a unsunpervised deep-learning method, where the generative network and the discriminator contest with each other. GAN-based methods do not need groundtruth and generate results with realistic characteristics.

From an information entropy perspective, GAN is a framework for estimating generative models through an adversarial process, which consists of two models: a generative model G that captures the distribution of the data, and a discriminative model D that estimates the probability that the samples came from the training data rather than G. The training procedure for G is to maximize the probability of D making a mistake. This framework corresponds to a minimax two-player game.

Although the existing deep-learning-based infrared and visible image-fusion methods perform well to some extent, these methods still suffer from several drawbacks. First, since IR and visible image fusion is an unsupervised task lacking labels for the fused images, existing fusion networks often directly use subjective losses [9]. Second, existing image-fusion networks ignore high-level semantic information and focus only on the fusion of global images. As a result, the local fusion results of the injected images are poor and the saliency of the targets is low [10].

Unlike image-fusion tasks, recognition tasks have a clear goal, such as a semantic-segmentation task, which relies on a large amount of labeled data [11]. The combination helps to improve the saliency and contrast of image fusion. Semantic-segmentation networks based on deep learning focus on mining the high-level semantic features of the image and restoring the resolution to the original image. The simple segmentation of the image also represents its target saliency. As a classical pixel-level task, semantic-segmentation techniques can extract semantics from the input data and play an important supporting role in many other tasks. The performance of many deep-learning-based tasks has been improved by exploiting the features of semantic segmentation. In image-fusion tasks, some methods claim to show that semantic segmentation is beneficial in guiding image fusion. However, these methods require obtaining segmentation labels as prior knowledge before image fusion, which is costly in terms of time and manpower during the testing phase.

To address the above issues, we propose Semantic Guided Visual Perception GAN (SGVPGAN), an adversarial framework for high-quality IR-visible image fusion. SGVPGAN is based on an Adversarial Semantic Guidance (ASG) and an Adversarial Visual Perception (AVP) module for visible-like and target-highlighted appearances. In addition, we constructed a joint distribution function between the fused images and the corresponding semantics and used the discriminator to improve the fusion performance of natural appearance and targets. Through qualitative subjective evaluation and quantitative objective evaluation, we find that SGVPGAN offers superior image-fusion performance compared to other infrared and visible image-fusion methods.

Our primary contributions are summarized as follows:We propose SGVPGAN, an adversarial framework for IR-visible image fusion, which has a visible-like and target-highlighted appearance.We develop the ASG and AVP for global and local enhancement during the training phase.The proposed SGVPGAN achieves significant improvements in terms of both visual appearance and quantification results.

## 2. Related Work

In this section, we briefly describe the background and related work, including generative adversarial networks, semantic segmentation, and the deep-learning-based fusion of infrared and visible images.

### 2.1. Generative Adversarial Network

The concept of generative adversarial networks was first introduced by Goodfellow et al. [12]. It has profound and wide applications in the field of image generation [13]. From the perspective of data distribution in image generation, GAN is an implicit generative model and the training-data distribution is contained in the trained generator G. Therefore, given any target domain A, a sample point similar to the distribution of domain A can be obtained through the GAN.

To achieve a better quality of generated images and a more stable adversarial training process, GAN has derived a series of variants during development. DCGAN replaces the multilayer perceptron (MLB) in the generator and discriminator of the original GAN with a convolutional neural network (CNN) for feature extraction. LSGAN uses least square loss instead of the cross-entropy loss of the initial GAN, which improves the performance of image generation and makes training more stable. CGAN adds additional conditions to the GAN, making the GAN generation process controllable [14]. WGAN introduces the Wasserstein distance and simply adapts the original GAN and achieves a surprising performance. To some extent, it solves the problems of pattern collapse, training difficulties, and instability, and the loss of generators can account for problems in the training process [15]. BigGAN improves the performance of GAN by increasing the number of parameters and scaling up the batch size. It uses the truncation technique to increase the stability of the training process. At the same time, it offers a certain balance between the stability of training and the performance of the network [16].

U-Net GAN is a recently developed GAN with excellent image-generation performance, which achieves state-of-the-art performance on several datasets [17]. Based on BigGAN, U-Net GAN changes the discriminator to a U-Net structure. The encoder discriminates the input image globally, while the decoder discriminates the image pixels locally. As a result, the generated images have higher quality and more realistic texture details.

The discriminator of our proposed network structure adopts the idea of U-Net GAN. We design the discriminator as a simple U-shaped network that classifies the global fusion results and improves the texture details of the fused images to some extent.

### 2.2. Semantic Segmentation

Semantic segmentation is a fundamental topic in computer vision, which recognizes targets at the pixel level. FCN was the first network that made a major breakthrough in the field of image segmentation using deep-learning methods. FCN replaced the fully connected layer of a neural network with a convolutional layer, thus creating a fully convolutional neural-network design. The U-Net structure [18] was proposed by Olaf et al. It was the first neural network for medical-image segmentation, and its network structure was more or less adopted by later semantic segmentation networks. Semantic-segmentation networks usually have a large number of parameters but often require real-time performance in their applications. Many semantic segmentation networks for real-time requirements have been proposed, such as ENet [19], ERFNet [20], ICNet [21], and BiseNet [22]. RPNet proposes a novel feature residual pyramid, which can help improve detail and edge information [23].

Since semantic segmentation not only extracts semantic features of images but also classifies images at the pixel level, it is often used in detection, tracking, and image-fusion tasks to provide target information and improve models. In detection tasks, DES uses segmentation modules to obtain attention masks and uses advanced semantic features to improve detection performance for small targets [24]. In the tracking task, Xue et al. use a segmentation network to obtain semantic information. The segmentation network provides an ROI mask to suppress background interference and improve the performance of the tracking network [25].

In an image-fusion task, the segmentation method preserves more semantic information for the image-fusion process. Hou et al. used the semantic-segmentation mask as prior knowledge for the image-fusion network. The mask divides the image into foreground and background and performs high-quality fusion so that the fused image retains more information [26]. SGVPGAN uses the semantic information extracted by the semantic segmentation network to guide the image-fusion network to generate fused images. Tang et al. cascade the image-fusion module and semantic-segmentation module and leverage the semantic loss to guide high-level semantic information to flow back to the image-fusion module, which effectively boosts the performance of high-level vision tasks on fused images [27]. Zhou et al. design an information quantity discrimination (IQD) block to guide fusion progress and identify the distribution of infrared and visible information in the fused image [28]. Wu et al. employ the semantic segmentation technology to obtain the semantic region of each object, and generate special weight maps for the infrared and visible image via pre-designed fusion weights [29].

### 2.3. Deep-Learning-Based Infrared and Visible Image Fusion

Deep-learning-based image-fusion methods mainly consist of supervised methods and GAN methods. Liu et al. used CNN to fuse IR and visible images, but they still used some traditional algorithms. The network is used to generate a weighted Gaussian pyramid and fuse each layer based on the weights. Deepfuse designed the network structure for image fusion, including feature-fusion and image-reconstruction modules. This network sets an objective human constraint that is determined by the SSIM loss between the fused image and the original image [30]. Densefuse improves the network structure by adding dense connections based on Deepfuse and adds L2 regularization to the SSMI loss [31]. Based on the physical prior, DRF innovatively decomposes the image into a physical property representation vector and a scene-representation feature map after being constrained by certain criteria and trained by the network. The fused image is reconstructed by fusing the two physical attributes [32].

Recently, GAN-based approaches have emerged and demonstrated significant potential. The FusionGAN proposed by Ma et al. is the first GAN-based network for fusing infrared and visible images [9]. As an end-to-end model, FusionGAN does not require the manual design of complex fusion rules. However, the infrared component of the fused images gradually diminishes during the training of FusionGAN. To address this problem, the authors propose DDcGAN with dual discriminators. Two discriminators can discriminate the fused image from the input infrared image and visible image, which enhances the infrared component of the fused image to some extent. Meanwhile, the authors tried to add detail loss and target enhancement loss to FusionGAN to solve the problem of the low saliency and coarse details of targets in FusionGAN [33]. Hou et al. proposed a semantic-segmentation-based fusion network for visible and infrared images [26]. The network adds image segmentation labels as prior knowledge to the network to utilize this information for image fusion. However, this also means that the labels need to be prepared in advance for testing.

Recently, a unified framework to solve different fusion problems, including multi-modal, multi-exposure, and multi-focus cases has became a research hot-spot. U2Fusion automatically estimates the importance of adaptive preservation degrees for effective and universal image fusion [34]. SwinFusion combines cross-domain long-range learning and Swin Transformer to integrate complementary information and global interaction [35]. GIDGAN designs a powerful gradient and intensity-discriminator generative adversarial network for gradient and intensity retention in image-fusion tasks [36].

In practical applications, labels for image segmentation are not always easy to obtain. Our proposed network architecture only uses labels as input during training, and does not require segmentation labels during testing. This avoids the extra work of manual labeling in real applications.

## 3. Method

### 3.1. Overall Network Architecture

The SGVPGAN framework consists of three components: a generator, a perceptual discriminator, and a semantic auxiliary module. The overall structure of SGVPGAN is shown in Figure 1. The RGB image $I_{vis}$ and the infrared image $I_{ir}$ are passed through two encoders, and the outputs of the two encoders are fused with the decoder of the single luminance channel of the image, which is used to reconstruct the fused RGB image $I_{f}$ with the color components of the original visible image $I_{vis}$. The perceptual discriminator is used to discriminate the true RGB image from the fused image to obtain a true fusion result, and the semantic auxiliary module is used to extract the semantics of the target and background from the fused image $I_{f}$ to guide the fusion network to generate a fused image with great target saliency. The generator and the discriminator establish an adversarial structure for SGVPGAN network training. In the testing phase, the generator fuses IR-visual image pairs to obtain realistic fused images without segmentation labels.

#### 3.1.1. Generator Network Architecture

The generator structure is shown in Figure 2. This generator adopts two encoders for both infrared and visual image extraction and a decoder for the feature fusion.

We feed the visible image $I_{vis}$ and the infrared image $I_{ir}$ into the dual encoder individually. These two encoders share the same network structure. Each convolution layer uses 3 × 3 convolution and the scale of the feature map is kept constant. In both decoders, the number of channels of the feature map is increased. There is no pooling layer during the process to prevent the loss of image information. Referring to DenseNet [37], we use dense concatenation to continuously replenish the information of the forward features. Dense concatenation ensures that the shallow features can be reused and efficiently utilized in deep convolution, which can effectively help our fused images retain more detailed information.

The output feature maps from the two encoders are fused in a cascaded manner and then fed to the decoder. The number of channels of the feature maps is gradually reduced during the decoder process. Finally, we use the sigmoid activation function to obtain a probability map with the number of channels as 2. The two channels are used as the probability distribution of the visible and infrared components in the fused image. The output feature maps of the two channels are points multiplied with the luminance channels of the visible and infrared images, respectively, and then are added together. After the tanh activation function, the fused single-channel image is produced. We confirmed through experiments that this operation can prevent the target from darkening as a function of the segmentation network. Throughout the architecture of the generator, we used spectral normalization (SN) [38], after each convolutional layer. To prevent gradient explosion or gradient disappearance and to speed up the convergence of the network, we added batch normalization (BN) to the network. We set Leaky Relu as the activation function. On the contrary, Relu loses negative values of the feature map during fusion, and information is lost for the fusion task. Leaky Relu can solve this problem.

#### 3.1.2. Perceptual Discriminator

The luminance channels of the visible image $I_{vis\_y}$ and the fused single image $I_{f\_y}$ are fed into the encoder in an unpaired manner to build a generative and adversarial structure. To obtain the global and local discrimination between the visible image and the fused image, and enhance the performance of the fused images on structure and textures, the perceptual discriminator employs a conditional U-shaped network with attention to detail and local structural information. The conditional discriminator utilizes the segmentation label as the condition in the discrimination of the visible image and the fused image to ensure the correspondence of the semantics. In addition, the U-shaped discriminator makes strong spatial discrimination [17]. The structure of the Perceptual discriminator is shown in Figure 3.

To construct the joint distribution function for perceptual and semantic enhancement, the input image and the corresponding segmentation label are concatenated together before being fed to the encoder. The label is an auxiliary condition. By adding segmentation labels, the discriminator can judge the fused image with higher quality based on this. It is helpful to optimize the details of the fused image, to discriminate the fused image spatially based on the high-level semantics in a reasonable way, and to impose certain constraints on the pixel-level fusion of the fused image. In other words, some high-level semantic information is sent to the U-discriminator, which drives the image fusion based on the semantic information and increases the amount of information in the fused image.

After being fed to the encoder, the number of channels of the feature map increases in full convolution. For each convolution, the size of the feature map is halved. The encoder extracts the global features throughout the process and, finally, obtains the global discriminant result through the global pooling layer and fully connected layer. Our global discriminant result is to discriminate the overall appearance of the visual image and the fused image. This improves the overall image perception of the fused image and makes it more natural.

In the decoder, we adopt transpose convolution for the output high-level feature map of the encoder. For each transpose convolution, the number of channels decreases while the size of the feature map doubles, creating an asymmetric relationship with the encoder. We continuously supplement the forward information with skipped connections, which effectively reuses the lost information in the encoder. After restoring the feature map to the original image size, we obtain a discriminant of the original image size by using convolution. This decision can be interpreted as a pixel-level decision in the image space. It allows to make a judgment on the local texture structure of the fused image and gives feedback to the generator in space. In the fusion task, this spatial decision can give more visibility to the local texture details of the fused image and improve the naturalness of the fused image from a local perspective.

After each layer of the discriminator, spectrum normalization is employed to increase the stability of the GAN in the training process. Similar to the generator, each layer uses batch normalization and Leaky Relu as the activation function.

#### 3.1.3. Semantic Auxiliary Module

We employ the RPNet segmentation network as a semantic auxiliary module. RPNet segmentation networks are based on residual pyramids, which have fewer parameters, fast inference and good segmentation performance. The color channel of the visible image is connected to the fused luminance channel, and then the YUV image is converted to an RGB image $I_{f}$ and fed into RPNet. Finally, we obtain a probability map and compute the segmentation loss with labels.

The segmentation network works as an auxiliary module. On the one hand, the network enhances the ability to mine the semantic features of the fused images. On the other hand, semantic information guides the generator to fuse images with better target saliency. The constraint of segmentation loss motivates the segmentation network to learn the semantics of the fused image, which, in turn, guides the fused image to make a suitable fusion in space to achieve high-quality image fusion. In contrast to the direct MSE loss between the fused and IR images, our semantic auxiliary module parses the fused images in a high-level manner to guide image fusion. This approach considers the spatial distribution of the fused images instead of the coarse global MSE loss.

### 3.2. Loss Function

The discriminator D and the generator G play the following two-player minimax game. It is expressed in the form of the sum of two minus cross-entropy losses (real samples and generated samples). This means that the discriminator maximizes the distribution gap between generated and training data, while the generator minimizes the distance between generated and training data.
(1)$$\;\min_{\theta (G)}\max_{\theta (D)}E_{x∼P_{data}}\log D\left(x\right)+E_{x∼P_{G}}\log\left(1-D\left(G(x)\right)\right)$$
where E expresses the expectation of the data distribution. The complete cross-entropy of the training sample is $1*\log D\left(x_{1}\right)+0*\log D\left(x_{0}\right)$, where $D\left(x_{0}\right)$ and $D\left(x_{1}\right)$ are the discriminator judgment probability (whether the training sample is a training sample (1) or a generated sample (0)). The full cross-entropy of the generated samples is $1*\log D\left(G\left(z_{1}\right)\right)+0*\log D\left(G\left(z_{0}\right)\right)$, where $D\left(G\left(z_{0}\right)\right)$ and $D\left(G\left(z_{1}\right)\right)$ represent the discriminator judgment probability (whether the generated sample is a training sample (1) or a generated sample (0)). $D\left(G\left(z\right)\right)$ needs to be as small as possible. However, in order to align (maximize) the direction of the cross-entropy function with the training sample, it is represented by log(1−D(G(x))).

In our network, there are three loss functions: visual perceptual loss (AVP loss), semantic guidance loss (ASG loss), and generator loss in the generative adversarial structure, which is used to train perceptual discriminator, semantic discriminator, and generator, respectively. The generator loss is composed of the perceptual adversarial loss, the semantic adversarial loss, and the detail loss.

#### 3.2.1. Visual Perceptual Loss

The perceptual discriminator is continuously enhanced to distinguish the visible image from the fused image during the training process and continuously gives feedback to the generator. The discriminator will discriminate it as true when the luminance channel $I_{vis\_y}$ of the visible image is input, and it will discriminate it as false when the fused image $I_{f\_y}$ is input as the fused image. The visual perceptual loss is conducted at the end of the encoder and decoder for structure and detail discrimination. We use $D^{U}$ to denote the discriminator, which consists of an encoder and a decoder, representing $D_{enc}^{U}$ and $D_{dec}^{U}$, respectively. The discriminator’s encoder outputs global information loss $L_{D_{enc}^{U}}$, and the discriminator’s decoder outputs local information loss $L_{D_{dec}^{U}}$. We use the following formula,
(2)$$\begin{array}{c} \;L_{D_{enc}^{U}}^{U}=-E_{I_{vis\_y}}\left[\min\left(0,-1+D_{enc}^{U}\left(I_{vis\_y}\right)\right)\right]-E_{I_{vis},I_{ir}}\left[\min\left(0,-1-D_{enc}^{U}\left(G\left(I_{vis},I_{ir}\right)\right)\right)\right] \end{array}$$
(3)$$\;L_{D_{dec}^{U}}^{U}=-E_{I_{vis\_y}}\left[\min\left(0,-1+D_{dec}^{U}\left(I_{vis\_y}\right)\right)\right]-E_{I_{vis},I_{ir}}\left[\min\left(0,-1-D_{dec}^{U}\left(G\left(I_{vis},I_{ir}\right)\right)\right)\right]$$
where ${\left[D_{dec}^{U}\left(I_{vis\_y}\right)\right]}_{i,j}$ and ${\left[D_{dec}^{U}\left(G\left(I_{vis},I_{ir}\right)\right)\right]}_{i,j}$ refer to the discriminator decision at pixel (*i,j*). E(·) indicates the mathematical expectation of the decision distribution. We follow the hinge loss proposed in U-Net GAN [17]. Our two loss functions represent global and local decision distances, respectively. Thus, in the process of the continuous strengthening of the discriminator, the discriminator cannot only make global decisions but also local decisions.

#### 3.2.2. Semantic Guidance Loss

The semantic guidance loss is based on the semantic auxiliary module. We adopt the RPNet segmentation network; however, we do not use the auxiliary loss in RPNet for simplicity. After inputting RPNet the fusion image $I_{f}$, we obtain the output segmentation result $I_{pred}$ from RPNet. Then we calculate cross-entropy loss between $I_{pred}$ and $I_{label}$. Both of them are normalized to [−1, 1]. The semantic guidance loss formula is shown as follows:(4)$$\;L_{s-adv}=L_{seg}=-\frac{1}{WH}\sum_{i,j}\sum_{c=0}^{N-1}y_{c}\left(i,j\right)log\left[p_{c}\left(i,j\right)\right]$$
where $y_{c}\left(i,j\right)$ denotes the value of channel *c* of $I_{label}$ one-hot vector at pixel (i,j). A one-hot vector here refers to the classification of whether a pixel is foreground or background, with foreground as vector [1, 0] and background as vector [0, 1]. $p_{c}\left(i,j\right)$ indicates the prediction value of the channel c of $I_{pred}$ at pixel (*i,j*). *N* refers to the number of channels. *W* and *H* refer to the width and height of the image.

#### 3.2.3. Generator Loss

The loss function of the generator consists of three main components: perceptual adversarial loss $L_{p-adv}$, semantic adversarial loss $L_{s-adv}$, and detail loss $L_{detail}$. The perceptual adversarial loss $L_{p-adv}$ is used to guide the fused image being recognized as real in the discrimination, and the global and local details of the fused image tend to be perceptual to visible light. The semantic adversarial loss $L_{s-adv}$ guides the fused image towards easy image segmentation. Since IR images contain richer semantic information, the semantic adversarial loss $L_{s-adv}$ is equivalent to adding target saliency information from IR images to the fused images, thus improving the target saliency of the fused images. The detail loss $L_{detail}$ is used to enhance the visible detail information of the fused image.

When computing the perceptual adversarial loss $L_{p-adv}$, the parameters of the discriminator are fixed and the parameters of the generator are trained. The purpose of the generator is to train the fused image that can cheat the discriminator. In addition, the perceptual adversarial loss $L_{p-adv}$ is calculated as follows:(5)$$\begin{array}{c} \;L_{p-adv}=-E_{I_{vis},I_{ir}}\left[D_{enc}^{U}\left(G\left(I_{vis},I_{ir}\right)\right)+{\left[D_{dec}^{U}\left(G\left(I_{vis},I_{ir}\right)\right)\right]}_{i,j}\right] \end{array}$$

When computing the semantic adversarial loss $L_{s-adv}$, the parameters of the semantic auxiliary module are fixed and the parameters of the generator are trained. The generator is continuously adjusted during the training process and gradually outputs fused images with better segmentation performance. The target saliency of the image will be improved and the effective information of the IR image will gradually increase in the fused image. The segmentation loss in the generator is also the cross-entropy loss. In FusionGAN, the content loss is the average difference between the pixel values of the fused image and the IR image. This results in the whole fused image becoming blurred due to too many infrared components. In contrast to this approach, we use adversarial segmentation loss to regionally increase the infrared component to make the target of the fused image more salient.

By using the adversarial loss to train the generator, the distance between the fused image and the visible light is gradually reduced so that the fused image has visible-light image perception. The output of encoder and decoder losses make the fused image globally and locally constrained by the visible image. Specifically, we follow the hinge loss used in the training of the U-Net GAN.

The detail loss $L_{detail}$ is the distance between the gradient of the fused image and the visible image. We calculate the gradient of both and find the average of the L2 criterion of the difference between the two gradients. Its formula is as follows.
(6)$$\;L_{detail}=\frac{1}{WH}\sum_{i,j}{\left(\nabla I_{f\_y}\left(i,j\right)-\nabla I_{vis\_y}\left(i,j\right)\right)}^{2}$$
where ∇ denotes the gradient of the image, (i,j) is the position of the pixel, *W* and *H* represent the width and height of the image, respectively. Detail loss is used to close the gradient of the fused image towards the gradient of visible light, which allows the fused image to have more detailed information.

Combining the three losses above, we obtain the loss function of the generator. The formula is as follows:(7)$$\;L_{G}=L_{p-adv}+\alpha L_{s-adv}+\varepsilon L_{detail}$$
where α and ε are hyperparameters balancing the weights of the three losses.

## 4. Experimental Results

### 4.1. Dataset

We evaluated our algorithm on the MF datase t [39]. The dataset is designed for semantic segmentation of visible and infrared images, which contains images of street scenes for autonomous vehicles. There are 1569 sets of RGB-T image pairs in the dataset, with 820 for daytime scenes and 749 for nighttime scenes. We trained the nighttime RGB-T dataset with 374 pairs of RGB-T images in the training set, 187 in the validation set, and 188 in the test set. The sizes of the images are 640×480. Although this dataset has been used as a dataset for semantic segmentation, the RGB-T images in this dataset are mostly pixel-aligned and, thus, can be used for image fusion. The dataset is also rich in night scenes, often with dim lights, glare, and other common driving scenes. Each pair of RGB-T images has a corresponding label. In addition to the unlabeled classes, the labels contain eight classes. Here, we use only three classes that are common in driving: cars, pedestrians, and bicycles. The pedestrian class is more prominent in the IR images.

### 4.2. Training and Test Details

In the training phase, data augmentation was adopted. We randomly cropped the images with a crop size of 400 × 400. Meanwhile, we used random shifting and random horizontal flipping for data augmentation, the detailed steps in the training phase is shown in Algorithm 1.
**Algorithm 1:** Training  SGVPGAN.
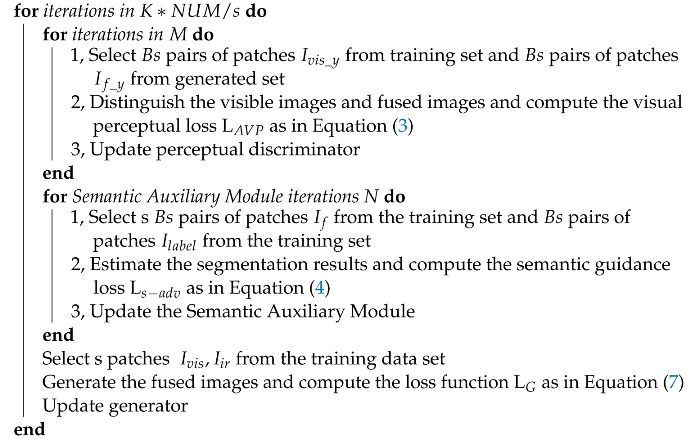


The tasks of GAN and semantic segmentation are very different. In practice, the semantic loss is easily converged, but the perceptual loss fluctuates as the generator and discriminator slowly converge in an adversarial way. That means that after several epochs, the perceptual loss will be much bigger than the semantic loss. Therefore, in the setting of the network loss hyperparameters, the semantic adversarial loss hyperparameter was set to 10,000 and the detail loss hyperparameter was set to 100. During the training phase, we set batchsize to Bs. For each epoch, we first trained the perceptual discriminator *M* times, then trained the semantic auxiliary module *N* times, and, finally, trained the generator once. The optimizer was Adam and the training epoch number was *K*. In our experiments, we set the parameters as Bs=4, M=2, N=2, and K=300, and the number of images in the training set to NUM=374. Regarding the loss parameters, we chose α = 10,000, ε = 100.

In the test phase, we kept only the generator. We removed the data-augmentation methods and input the RGB-T test images with the original size to obtain the fusion result. The NVIDIA TITAN RTX and 32 GB of memory were used for training and testing.

### 4.3. Evaluation Metrics

Based on the benchmark VIFB [40] for visible and infrared image fusion, we conclude that the various evaluations of image fusion can hardly reflect the quality of image fusion, indeed; however, some objective evaluations can reflect the quality of image fusion to some extent. Therefore, we will demonstrate the superiority of our fusion method by both objective and subjective evaluations in this section.

#### 4.3.1. Objective Evaluation

We used the following metrics to evaluate the performance: AG (average gradient), EI (edge intensity), SF (spatial frequency), EN (entropy), mutual information (MI), and multi-exposure fusion structural similarity (MEF-SSIM) [41]. These evaluations can quantitatively evaluate our image quality based on image features and information theory, and can comprehensively evaluate our image-fusion quality. Our fusion image is based on target saliency, so we compared our image quality with other methods under the mask of segmentation labels.

AG (average gradient) is employed to measure the clarity of the fused image. The greater the average gradient, the better the image clarity and the better the quality of image fusion [42]. The formula of AG is as follows:(8)$$\begin{array}{cc} \;AG= & \frac{1}{\left(M-1)(N-1\right)}\sum_{i=1}^{M-1}\sum_{j=1}^{N-1}\\ \; & \sqrt{\frac{{\left(I_{f}\left(i+1,j\right)-I_{f}\left(i,j\right)\right)}^{2}+{\left(I_{f}\left(i,j+1\right)-I_{f}\left(i,j\right)\right)}^{2}}{2}} \end{array}$$
where *M* and *N* denote the width and height of the image, respectively.

EI (edge intensity) calculates the edge intensity of the fused image. Similarly, there is no need for the original reference images. The greater the edge intensity, the better the quality of the fused image [43]. The formula of AG is shown as follows: (9)$$\begin{array}{c} \;EI\left(I_{f}\right)=\frac{\sqrt{\sum_{i=1}^{M}\sum_{j=1}^{N}(s_{x}{\left(i,j\right)}^{2}+s_{y}{\left(i,j\right)}^{2})}}{MN}\\ \;s_{x}=I_{f}*h_{x},s_{y}=I_{f}*h_{y} \end{array}$$
where $\;h_{x}$ and $\;h_{y}$ are the Sobel operator in *x* and *y* directions. $\;s_{x}$ and $\;s_{y}$ are the results after employing Sobel operator to the fused image.

SF (spatial frequency) was used to calculate the change rate of the image gray level. The larger the spatial frequency, the clearer the image and the better the fusion effect [44]. The formula of AG is as follows:(10)$$\begin{array}{c} SF=\sqrt{RF^{2}+CF^{2}}\\ RF=\sqrt{\frac{1}{MN}\sum_{i=1}^{M}\sum_{j=2}^{N}{\left(I(i,j)-I(i,j-1)\right)}^{2}}\\ CF=\sqrt{\frac{1}{MN}\sum_{i=2}^{M}\sum_{j=1}^{N}{\left(I(i,j)-I(i-1,j)\right)}^{2}} \end{array}$$
The larger the SF, the higher the image quality.

EN (entropy) calculates the amount of information contained in the image. A larger value indicates more information in the image, indicating more details [45]. The formula of EN is as follows: where $p_{I_{f}}\left(i\right)$ is the statistical probability of the gray histogram.
(11)$$EN\left(I_{f}\right)=\sum_{i=0}^{255}-p_{I_{f}}\left(i\right)log_{2}\left(p_{I_{f}}\right)$$

MI measures the degree of dependency between visible and infrared images; the calculation method refers to [46]. MEF-SSIM is an advanced MEF image-quality model, and the calculation method refers to [47].

We chose the typical traditional methods and deep-learning methods developed recently to compare the performance, including ADF [48], CNN [49], DLF [50], FPDE [8], GFF [6], Hybrid-MSD [49], MGFF [51], MST-SR [52], ResNet [53], RP-SR [54] TIF [55], VSMWLS [56] and FusionGAN [9]. The results are shown in Table 1:

The above methods include excellent traditional methods and deep-learning methods in recent years. Bold and underlined numbers indicate that the method ranks first in the fusion evaluation index, and only bold numbers indicate that the method ranks second in the fusion evaluation index. From the comparison of these objective evaluations, our fusion method has advantages over these methods.

As the measures used mainly evaluate the richness of textures, they do not directly indicate the fusion performance, e.g., the noise in the dark visible images will interfere with the quantitative evaluation. The proposed method highlights the target in the fusion result and maintains the textures at the same time. In addition, the inference time per image of the proposed method is competitive.

#### 4.3.2. Subjective Evaluation

As there are many evaluations for image-fusion tasks, it is difficult to evaluate them in a uniform manner when comparing various methods in practice. Furthermore, although they are highly evaluated quantitatively, the perception of image quality may not be satisfactory. Therefore, here, we evaluate the fused images qualitatively and compare our fused images with other fusion methods.

As can be seen from the sample 1 in Figure 4, our image basically preserves the details of the billboard of the visible image. In addition, the infrared radiation details of the building can be retained in our image. However, other methods retain less such information. More importantly, it is difficult for other methods to observe the faint IR information of distant pedestrians, while in our method we can see the distant people more clearly, which proves that pedestrians with distinct IR radiation features will be more salient in our fusion method.

From the sample 2 in Figure 4, we can see that our method still retains the details such as leaves in the visible image and the infrared details of buildings next to leaves are also evident. The bicycles with more salient detail information in the infrared image can also be displayed in the fusion image. Furthermore, our method clearly retains human information and has a high contrast ratio. Other methods clearly lose target saliency information in the infrared image.

The sample 3 in Figure 4 shows a strong light scene at night. In visible light, pedestrians, cars and other targets cannot be seen due to the presence of glare. In our fusion image, the car and the person in the car are more salient than in other fusion methods, and the person on the right is also more salient. In other fusion images, it is almost difficult to distinguish the car, the person in the car, and the person on the right. It can be seen that the significant advantage of our target prominence in the fused image is obvious in the strong light scene. On the contrary, in other methods, the target in the glare scene loses some infrared information, and less target information is retained, and it is even difficult to identify the target.

### 4.4. Ablation Study

#### 4.4.1. U-Shaped Discriminator Experiment

We design a U-shaped discriminator in the AVP module to distinguish the fused image globally and locally, so as to enhance the local detail and perception of the image. To validate the effectiveness of our method, we removed the decoder of the U-shaped discriminator under the condition that the overall framework of the original network remains unchanged.

The comparison of the objective evaluation results of the U-shaped ablation experiment is as in Table 2 and the subjective evaluation results is as in Figure 5.

After the U-shaped discriminator is added, the objective indicators AG and EI are improved, which means that the fused image contains more information and the image fusion effect is improved.

From Figure 5, we observe that our fused image offers more visible detail information when using a U-shaped discriminator. For example, the details of the billboard and light in the visible image are well-retained when we use the U-shaped discriminator. This shows that our U-shaped discriminator enhances the visible detail of fused images and the visible perception of the image to a certain extent. Obviously, the detail of the pedestrian is also rich with a U-shaped discriminator.

#### 4.4.2. Conditional Discriminator Experiment

At the input of the perceptual, we also input the segmentation label as the prior knowledge of the discriminator, so that the discriminator can obtain the target prior information. We maDe the network automatically select the fusion rules spatially. The discriminator has the ability to perceive high-level semantics and spatial information. In order to demonstrate the effectiveness of our method, we conducted comparison experiments. The results are shown in Table 3 and Figure 6:

After the conditional discriminator is added, the objective indicators AG, EI, SF, and EN are improved, which means that the fused image contains more information and the image fusion effect is improved. From Figure 6, it is clear to see that the contrast of the ground is significantly enhanced and the detail information is richer when the conditional discriminator is used. This indicates that the segmentation labels added as a prior knowledge can optimize the spatial distribution and improve the perceptual capability of the fusion network.

#### 4.4.3. Semantic Auxiliary Module Experiment

To show the role of the semantic auxiliary module, we adjust the hyperparameters of the semantic adversarial loss of the generators to show the effect of the semantic auxiliary module on the fused images.

We adjusted the hyperparameters of the semantic adversarial loss to 1000, 5000, 10,000, and 20,000. The comparison of the objective evaluation results of the different losses experiment is as in Table 4 and the objective evaluation results are as in Figure 7.

It is obvious that when the hyperparameters are small, there are more visible details in the fused images, but the targets with distinct IR thermal features are significantly blurred, which means less IR salient information. As α gradually increases, the semantics of the segmentation network gradually guides the progress of fusion, and more IR image components containing salient semantic information are added to the fused image. However, when α is larger, the detail loss of the visible image is larger, and more infrared information makes the fused image blurred. Therefore, we need to reasonably adjust the hyperparameter α to keep a balance between the detailed information of the visible image and the prominent information of the infrared image. Based on the experimental results, we choose α = 10,000 to balance this relationship.

## 5. Conclusions

In this paper, we propose a novel infrared and visible image-fusion framework SGVPGAN for night vision enhancement, which has a visible-like and target-highlighting appearance. The pixel-level ASG and AVP in the framework combine semantic segmentation and visual enhancement, and significantly improve the fused images. Experiments on the MF dataset demonstrate that our fusion method outperforms other popular fusion methods with qualitative and quantitative evaluations. In the future, we will optimize our network to achieve better performance and improve the training strategy of the GAN architecture. The proposed method has a wide range of applications in intelligent vehicles and great potential in scene perception.

## Figures and Tables

**Figure 1 entropy-24-01327-f001:**
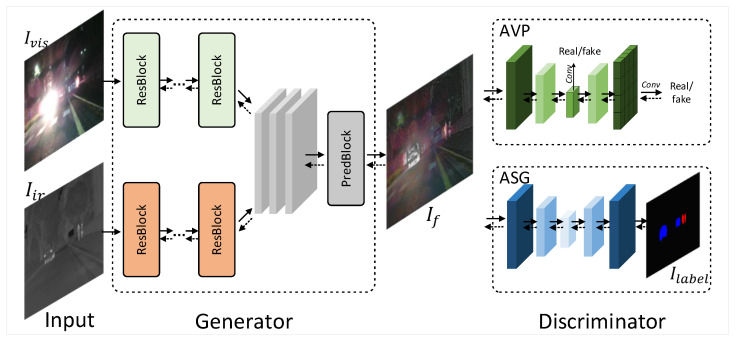
SGVPGAN network architecture. Solid arrows indicate the forward pass and dashed arrows indicate the backward pass.

**Figure 2 entropy-24-01327-f002:**
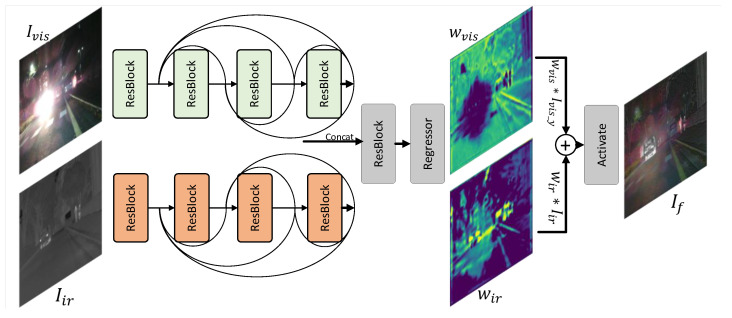
Inference flowchart of the generator.

**Figure 3 entropy-24-01327-f003:**
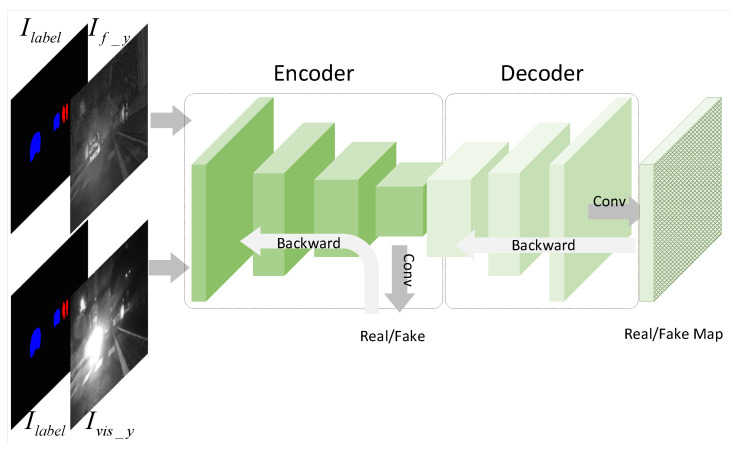
The structure of the perceptual discriminator.

**Figure 4 entropy-24-01327-f004:**
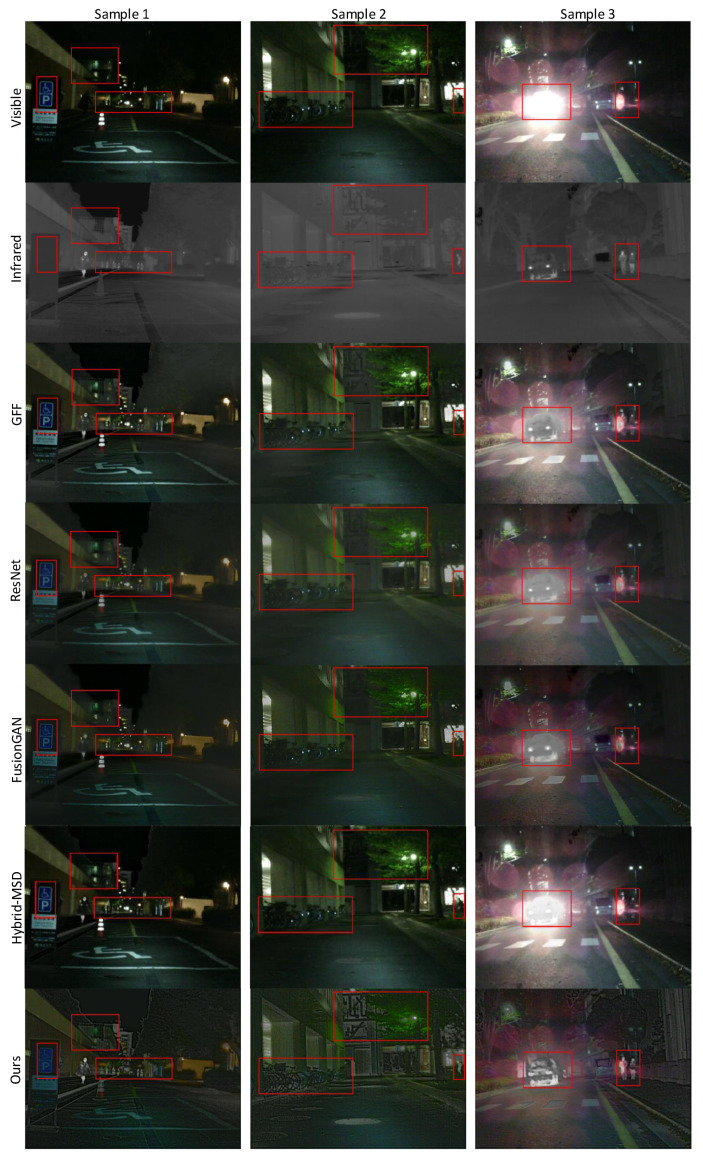
The sample image of subjective experiments of various methods on the MF dataset.

**Figure 5 entropy-24-01327-f005:**
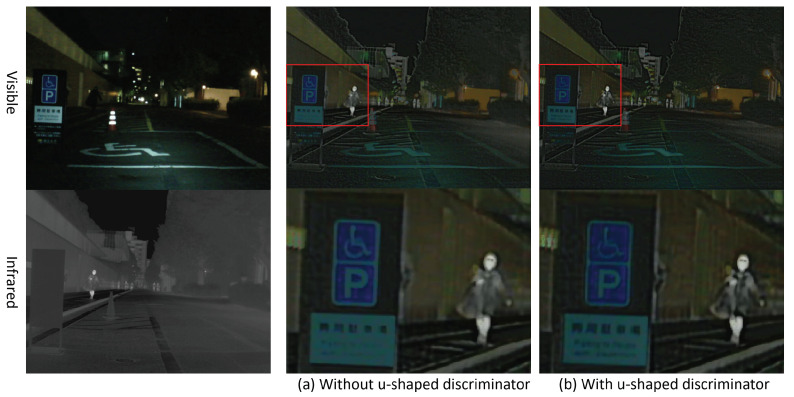
Experiment on ablation of the U-shaped discriminator.

**Figure 6 entropy-24-01327-f006:**
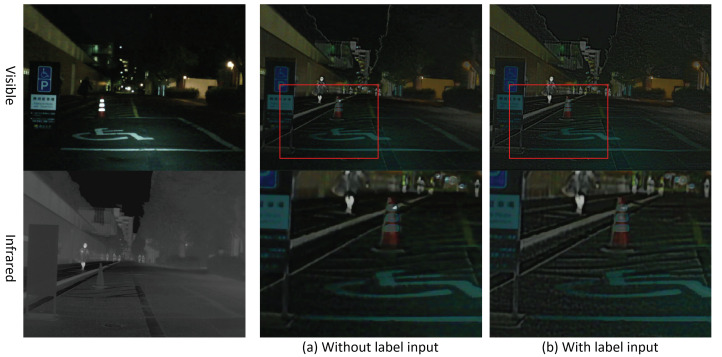
Experiment on ablation of the segmentation label.

**Figure 7 entropy-24-01327-f007:**
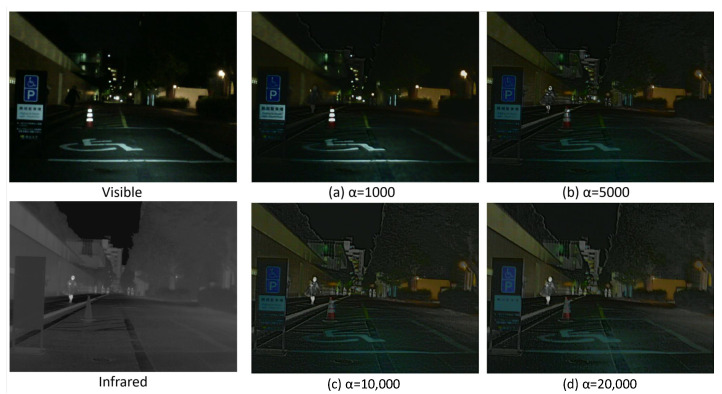
Experiment on the hyperparameters of the semantic auxiliary module.

**Table 1 entropy-24-01327-t001:** Objective evaluation of fusion algorithms. The bold-underlined indicates the best, and the bold indicates the second best.

	AG	EI	SF	EN	MI	MEF-SSIM	Inference Time (s)
ADF	0.166	1.77	2.58	0.378	1.70	0.919	0.619
CNN	0.207	2.22	3.05	0.382	**2.17**	0.904	6.448
DLF	0.162	1.74	2.55	0.377	1.78	**0.932**	6.726
FPDE	0.169	1.81	2.59	0.379	1.70	0.915	1.450
GFF	0.213	2.28	3.11	0.392	1.64	0.907	0.393
Hybrid-MSD	**0.227**	**2.42**	**3.26**	0.391	**2.27**	0.904	4.929
MGFF	0.219	2.34	3.11	0.385	1.45	0.902	1.455
MST-SR	0.209	2.24	3.05	0.386	2.14	0.903	0.709
ResNet	0.162	1.74	2.54	0.378	1.71	0.932	0.532
RP-SR	0.200	2.13	3.07	**0.394**	1.60	0.905	0.369
TIF	0.216	2.32	3.14	0.388	1.52	0.904	**0.061**
VSMWLS	0.222	2.36	3.19	0.387	1.78	0.905	2.351
FusionGAN	0.170	1.83	2.67	0.377	1.49	0.914	0.369
Ours	**0.249**	**2.69**	**3.25**	**0.394**	1.95	**0.922**	**0.121**

**Table 2 entropy-24-01327-t002:** Objective evaluation results of the U-shaped discriminator ablation experiment. The bold indicate the best.

	AG	EI
Without a U-shaped discriminator	0.236	2.553
With a U-shaped discriminator	**0.248**	**2.689**

**Table 3 entropy-24-01327-t003:** Objective evaluation results of the conditional discriminator ablation experiment. The bold indicate the best.

	AG	EI	SF	EN
Without a segmentation label	0.217	2.351	0.389	3.106
With a segmentation label	**0.248**	**2.689**	**0.394**	**3.246**

**Table 4 entropy-24-01327-t004:** Objective evaluation results of different hyperparameters of the semantic adversarial loss experiment. The bold indicate the best.

	AG	EI	SF	EN
α = 1000	0.139	1.504	0.365	2.246
α = 5000	0.198	2.128	0.379	2.700
α = 10,000	**0.248**	**2.689**	0.394	**3.246**
α = 20,000	0.235	2.540	**0.395**	3.243

## Data Availability

Publicly available datasets were analyzed in this study. This data can be found here: https://www.mi.t.u-tokyo.ac.jp/static/projects/mil_multispectral/, accessed on 18 September 2022.

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
