# Peer review of "Infrared-Visible Image Fusion Based on Semantic Guidance and Visual Perception"

_entropy, 2022, doi:10.3390/e24101327_

Round 1

Reviewer 1 Report

This paper presents a deep-learning approach to the fusion of an RGB image and an infrared one using segmentation to drive the fusion. The visual results look interesting and, according to the four quantitative measures presented, they improve on the state of the art. With some amount of editing, this paper might be acceptable for an image processing/deep learning venue. I’m not sure why the authors decided to submit it to Entropy, and on this issue below are my two general comments on the paper:

 On the one hand, the way the paper is presented might be acceptable in an image processing/deep learning venue, where one can count on the fact that the prospective audience are aware of the context of the paper and thus much background explanation could be skipped, but that is not the case in Entropy venue. Thus, much more top-down approach should be used, Figures should be more explanatory, captions should be self-explanatory, and equations for the loss functions should be more explained. Also, some background on adversarial networks would be helpful.

On the other hand, I do not see too much interest to Entropy audience unless the authors explain/argue what’s the advance (no problem if it’s only incremental) in information theory applications. You can argue that information measures as entropy and cross entropy are used in the paper, entropy as a measure of the scene information, and cross entropy as a loss function. But those measures have already been used for these purposes frequently , thus I do not see anything additional to be learned from the paper. In addition, the equations for cross entropy are confusing and it is not clear to me that they are used with probability distributions.

As I do not think that these two points, presentation and lack of focus, can be solved in the short time review cycle of Entropy journal, I recommend reject and resubmission once these questions (and the additional comments below) are addressed. Alternatively, the authors could resubmit to some image processing venue, such as mdpi Journal of Imaging.

Additional comments:

-Figure 1 should be more complete, for instance feedback from discriminator to generator does not appear. The input images are not labeled. In Figure 2 the weights are multiplied by the full images, while in text it explains that they are by the Y (luminance) channel. The whole process (channel separation and then merging result of generation with color channels) should be clear in the Figure(s).

-No comparison in time cost is given between the different methods. I understood that the cost is similar, but if so, it should be explained, and if not, costs should be compared.

-Conditional U-shaped network: please give a short explanation and/or reference.

-Equation 4: If it is meant to be a cross entropy it lacks a minus sign. Also, y_c and p_c should be probability distributions, are they?. What do they represent? Pixel luminance?.

-Equations 2,3,5: What do “E” stand for? I suppose expected value, but you have to mention it. Which are the distributions? What is T in RGBT? The label? Which values are assigned to labels? Is RGTB the joint distributions between image and semantics?.

-Equations 2,3: Why the difference between formula 2 and 3? Are each term of Formulae 2 and 3 meant to be a cross-entropy? And in that case, do the bracketed expressions represent distributions?.

-Equation 5: Why there is no expectation at the second term in Equation 5?. Without expectation, it can be very large or even infinite if log is applied to a very small value or to zero.

-As it is very difficult to difference results according to visual comparison, and I guess that the visual evaluation might be based on the (possibly different) objectives of fusion, maybe the authors, to strengthen more their point, could include some other measure in the comparison, such as mutual information and MEF-SSIM.

-In the compared images, I would like to see the nearest scoring quantitative method to the presented one, it seems to be the hybrid-MSD.

-If hyperparameter \alpha is given so a high value, much bigger than \epsilon (=100) and than the value corresponding to L_p-adv (=1), doesn't imply it that in practice the semantic loss L_s-adv is the only player? If not, why there is this big difference in orders of magnitude between the different losses?.

-I could not find the equation for L_s-adv.

-There is a duplicated paragraph: “In the test phase…”

Author Response

#Reviewer 1

Main comments:

(1) On the one hand, the way the paper is presented might be acceptable in an image processing/deep learning venue, where one can count on the fact that the prospective audience are aware of the context of the paper and thus much background explanation could be skipped, but that is not the case in Entropy venue. Thus, much more top-down approach should be used, Figures should be more explanatory, captions should be self-explanatory, and equations for the loss functions should be more explained. Also, some background on adversarial networks would be helpful.

Response: Thanks for your comments. We have improved the figures' illustrations and the captions. More explanation has been added for the loss function. We also added some background on adversarial networks in Section 1.

(2) On the other hand, I do not see too much interest to Entropy audience unless the authors explain/argue what’s the advance (no problem if it’s only incremental) in information theory applications. You can argue that information measures as entropy and cross entropy are used in the paper, entropy as a measure of the scene information, and cross entropy as a loss function. But those measures have already been used for these purposes frequently, thus I do not see anything additional to be learned from the paper. In addition, the equations for cross entropy are confusing and it is not clear to me that they are used with probability distributions.

Response: In the task of visible-infrared fusion, multi-modal data are analyzed and processed to generate fused images that have more perceptual information than source modal images. In this paper, we propose an adversarial framework for image fusion, that combines semantic analysis and Generative Adversarial Networks (GAN) to generate visible-like and target-highlighted appearances. This is a novel application of information theory and has more visual advantages than other image fusion methods.

Additional comments:

(3) Figure 1 should be more complete, for instance feedback from discriminator to a generator does not appear. The input images are not labeled. In Figure 2 the weights are multiplied by the full images, while in the text it explains that they are by the Y (luminance) channel. The whole process (channel separation and then merging result of generation with color channels) should be clear in the Figure(s).

Response: Thanks for your comments, we have improved Figure 1 and corrected the mistake in Figure 2.

(4) No comparison in time cost is given between the different methods. I understood that the cost is similar, but if so, it should be explained, and if not, costs should be compared.

Responses: Thanks for your comments. Inference time comparison has been added in Table 1.

(5) Conditional U-shaped network: please give a short explanation and/or reference.

Response:

Responses: The explanation has been added to section 2.1.2

To obtain the global and local discrimination between the visible image and the fused image, and enhance the performance of the fused images on structure and textures, the perceptual discriminator employs a conditional U-shaped network with attention to detail and local structural information. The conditional discriminator utilizes the segmentation label as the condition in the discrimination of the visible image and the fused image to ensure the correspondence of the semantics. Besides, the U-shaped discriminator makes strong spatial discrimination.

(6) Equation 4: If it is meant to be a cross-entropy it lacks a minus sign. Also, y_c and p_c should be probability distributions, are they? What do they represent? Pixel luminance.

Response: Sorry, “Then we calculate cross-entropy loss between If and Ilabel.” should be “Then we calculate cross-entropy loss between Ipred and Ilabel.”.

Ipred is the segmentation results and the Ilabel is ground truths. yc and pc are one-hot vectors of Ipred and Ilabel, which are probability distributions (segmentation results, not luminance). In Equation 4, we missed the minus sign, and have corrected mistakes.

(7) Equations 2,3,5: What do “E” stand for? I suppose expected value, but you have to mention it. Which are the distributions? What is T in RGBT? The label? Which values are assigned to labels? Is RGTB the joint distributions between image and semantics?.

Response: The “E” stands for mathematical expectation and discriminator decision that the If_y and Ivis_y are real or fake. we have added the descriptions in section 2.2.1.

We used RGBT to indicate Ivis and Iir, and have corrected that. The labels a are only used in semantic guidance loss, and it is unsupervised in Equations 2,3,5.

(8) Equations 2,3: Why the difference between formula 2 and 3? Are each term of Formulae 2 and 3 meant to be a cross-entropy? And in that case, do the bracketed expressions represent distributions?.

Response: The bracketed expressions represent distributions that the If_y and Ivis_y are real or fake. Formulae 2 and 3 are meant to be a cross-entropy. The Visual perceptual loss is conducted at the end of the encoder and decoder for structure and detail discrimination. So Formulae 2 and 3 respectively indicate the discrimination at the encoder and decoder.

(9) Equation 5: Why there is no expectation at the second term in Equation 5?. Without expectation, it can be very large or even infinite if log is applied to a very small value or to zero.

Response: Sorry, we missed a bracket in Equation 5, and have corrected that.

(10) As it is very difficult to difference results according to visual comparison, and I guess that the visual evaluation might be based on the (possibly different) objectives of fusion, maybe the authors, to strengthen more their point, could include some other measure in the comparison, such as mutual information and MEF-SSIM.

Response: We have added the Mutual Information (MI) and MEF-SSIM comparison in Table 1. As the purpose of the proposed method is to make the fusion results more informative and suitable for visual perception, our method does not get a high score of MI and MEF-SSIM,  but has advantages on the target highlighting.  

(11) In the compared images, I would like to see the nearest scoring quantitative method to the presented one, it seems to be the hybrid-MSD.

Response: We have added compared images of Hybrid-MSD.

As the measures used mainly evaluate the richness of textures, and do not directly indicate the fusion performance, e.g. the noise in the dark visible images will interfere with the quantitative evaluation. The proposed method highlights the target in the fusion result and maintains the textures at the same time. The related discussion has been added in section 3.3.1.

(12) If hyperparameter \alpha is given so a high value, much bigger than \epsilon (=100) and than the value corresponding to L_p-adv (=1), doesn't imply it that in practice the semantic loss L_s-adv is the only player? If not, why there is this big difference in orders of magnitude between the different losses?.

Response: The tasks of GAN and semantic segmentation are very different. In practice, the semantic loss is easily converged, but the perceptual loss is fluctuating as the generator and discriminator slowly converge in an adversarial way. That means after several epochs, the perceptual loss will be much bigger than semantic loss. To make a balance in the training period, we set the hyperparameter like this.

Besides, in the ablation study of perceptual loss, the effects of the perceptual module are also demonstrated. The related discussion has been added in section 3.2.

(13) I could not find the equation for L_s-adv.

Response: Sorry, The Ls-adv and Lasg are both represent Semantic Guidance Loss. We have corrected that and uniformly revised as Ls-adv.

(14) There is a duplicated paragraph: “In the test phase…”

Response: Sorry, we have corrected that.

Reviewer 2 Report

This paper proposed Adversarial Semantic Guidance and Adversarial Visual Perception modules to guide the infrared-visible image fusion.

g. Still, I think some problems need further investigation before it can be published.

1. I'm afraid I have to disagree that the proposed method is first combining image fusion and semantic segmentation network. Some existing works have tried to use semantic information to promote fusion performance. 

[1] Semantic guided infrared and visible image fusion. IEICE Transactions on Fundamentals of Electronics, Communications and Computer Sciences. 2021.

[2] Semantic-supervised Infrared and Visible Image Fusion via a Dual-discriminator Generative Adversarial Network. IEEE TMM,2021.

[3] Image fusion in the loop of high-level vision tasks: A semantic-aware real-time infrared and visible image fusion network. Information Fusion, 2022.

2. The authors are suggested to give more details in the Figures and their captions.

3. In my understanding, the ASG module is an auxiliary task to help the fusion task to extract more semantic information. Why do you name it a discriminator?

4. What is the role of the "Regressor" in Figure 2? Please give more information.

5. The descriptions in $2.2.1-2.1.4 are hard to read. The authors are suggested to draw a Figure for each module and provide corresponding explanations to help the readers understand how they are beneficial to the fusion task.

6. Some highly related fusion methods could be mentioned and commented on, including:

[1] U2Fusion: A Unified Unsupervised Image Fusion Network. IEEE TPAMI, 2022.

[2] SwinFusion: Cross-domain Long-range Learning for General Image Fusion via Swin Transformer. IEEE/CAA JAS, 2022.

[3] Unified gradient-and intensity-discriminator generative adversarial network for image fusion. Information Fusion, 2022.

7. Some typos need to be corrected, such as in P6(Line 191: DenseNetcite --> DenseNet)

Author Response

#Reviewer 2

This paper proposed Adversarial Semantic Guidance and Adversarial Visual Perception modules to guide the infrared-visible image fusion.

  1. Still, I think some problems need further investigation before it can be published

(1) I'm afraid I have to disagree that the proposed method is first combining image fusion and semantic segmentation network. Some existing works have tried to use semantic information to promote fusion performance.

[1] Semantic guided infrared and visible image fusion. IEICE Transactions on Fundamentals of Electronics, Communications and Computer Sciences. 2021.

[2] Semantic-supervised Infrared and Visible Image Fusion via a Dual-discriminator Generative Adversarial Network. IEEE TMM,2021.

[3] Image fusion in the loop of high-level vision tasks: A semantic-aware real-time infrared and visible image fusion network. Information Fusion, 2022.

Response: Sorry for insufficient investigation, we have deleted the sentence.

(2) The authors are suggested to give more details in the Figures and their captions.

Response: Thanks for your comments, we have added more details in Figures 1, and 2 and their captions.

(3) In my understanding, the ASG module is an auxiliary task to help the fusion task to extract more semantic information. Why do you name it a discriminator?

Response: Sorry for the misconception. We have revised the full text and corrected the ASG module to "Semantic Auxiliary Module".

(4) What is the role of the "Regressor" in Figure 2? Please give more information.

Response: “Regressor” indicate the Convolution operator that outputs the  and . We have changed the “Regressor” to “Convolution” in Figure 2.

(5) The descriptions in $2.2.1-2.1.4 are hard to read. The authors are suggested to draw a Figure for each module and provide corresponding explanations to help the readers understand how they are beneficial to the fusion task.

Response: Figure 3 has been added to demonstrate the structure of the Perceptual discriminator. Figure 2 is used to demonstrate the inference flowchart of the Generator. In this paper, we employ the semantic segmentation network RPNet as the Semantic Auxiliary Module, so the network architecture is not shown, but the illustration of the effect can be found in Figure 1.

(6) Some highly related fusion methods could be mentioned and commented on, including:

[1] U2Fusion: A Unified Unsupervised Image Fusion Network. IEEE TPAMI, 2022.

[2] SwinFusion: Cross-domain Long-range Learning for General Image Fusion via Swin Transformer. IEEE/CAA JAS, 2022.

[3] Unified gradient-and intensity-discriminator generative adversarial network for image fusion. Information Fusion, 2022.

Response: We have added discusses the mentioned methods in Section 2.

(7) Some typos need to be corrected, such as in P6(Line 191: DenseNetcite --> DenseNet)

Response: Thanks for your comments, we have corrected the mistakes and carefully revised the manuscript.

Round 2

Reviewer 1 Report

The authors improved the paper, corrected typos and added comparisons with new measures, and very importantly, they gave time cost. The method scores best or second best in most objective measures, it is fast and it seems to obtain good discriminative results. Thus, I’m happy about it. But many things remain. I’m still thinking that this paper is more fitted for an image processing/deep learning venue, as the amount of background needed is not at all common for most of Entropy community. For instance, to my request on introducing GAN the authors added a sentence where they use the “discriminator” concept, which is part of GAN, without having introduced this concept previously. It also still lacks a clear link with information theory. I understand this link is the use of cross entropy as loss function, but it is not clear to me whether some of the given loss functions are or not cross entropies, neither if the distributions used are normalized or not. The authors should be clear about it. And I think the use of a table with the definitions of all variables and distributions would help the reader a lot.

More detailed comments follow:

-Formula 3: Is it Cross entropy? Ivis, Ir: What distribution is it?

-Formula 4: Are distributions normalized?

-What is a hot-vector?

-Formula 5, is it a sum of cross entropies?

 About the last sentence in 3.3.1: “As the purpose of the proposed method is to make the fusion results more informative and suitable for visual perception, our method does not get a high score in MI and MEF-SSIM but has advantages in the target highlighting.”

I don’t understand this sentence. It seems as if MI and MEF-SSIM would be inversely correlated to "more informative and suitable for visual perception", which I do not think is the case. The authors have not to be afraid of not being first in all measures. No method, in fusion in general, is better than all the other ones in all measures. Thus I think the authors can simply drop this sentence.

Author Response

Question1:

For instance, to my request on introducing GAN the authors added a sentence where they use the “discriminator” concept, which is part of GAN, without having introduced this concept previously. It also still lacks a clear link with information theory. I understand this link is the use of cross entropy as loss function, but it is not clear to me whether some of the given loss functions are or not cross entropies, neither if the distributions used are normalized or not. The authors should be clear about it. And I think the use of a table with the definitions of all variables and distributions would help the reader a lot.

Response1:

Thank you for your suggestions. We added more explanation to GAN in the Introduction, Related Work section and the loss function part.

From an information entropy perspective, GAN is a framework for estimating generative models through an adversarial process, which consists of two models: a generative model G that captures the distribution of the data, and a discriminative model D that estimates the probability that the samples came from the training data rather than G. The training procedure for G is to maximize the probability of D making a mistake. This framework corresponds to a minimax two-player game. In the space of arbitrary functions G and D, a unique solution exists, with G recovering the training data distribution and D equal to 1/2 everywhere.

For variables distribution, GAN is an implicit generative model and the training data distribution is contained in the trained generator G.

The Discriminator D and the generator G play the following two-player minimax game. It is expressed in the form of the sum of two cross-entropy loss (real samples and generated samples). This means that the discriminator maximizes the distribution gap between generated and training data, while the generator minimizes the distance between generated and training data.

Question2:

Formula 3: Is it Cross entropy? Ivis, Ir: What distribution is it?

Response:

This is Hinge loss, which increases the training efficiency compared with the original GAN loss by maximizing the difference between the real and fake possibility.

Ivis and Ir represent the visible and infrared images, and their distribution is implicit in the dataset. If (fusion images) are distributed based on the infrared images Ir and the visible images Ivis, which has visible-like and target-highlighting appearance.  

Question3:

Formula 4: Are distributions normalized?

Response:

The data distribution has been normalized, and we processed the image colors from a distribution of [0, 255] to normalized distribution of [-1, 1]. We have supplemented this.

Question4:

What is a one-hotvector?

Response:

 We have added relevant explanations. In a general sense, a one-hot vector refers to a vector in which only one bit is 0 and the other bits are 1. In our work, one-hot vector refers to the classification of whether a pixel is a foreground or background, with foreground as [1, 0] and background as [0, 1].

Question5:

Formula 5, is it a sum of cross entropies?

Response:

Yes, it is a sum of cross entropies of the training samples and the generated samples.

Question6:

About the last sentence in 3.3.1: “As the purpose of the proposed method is to make the fusion results more informative and suitable for visual perception, our method does not get a high score in MI and MEF-SSIM but has advantages in the target highlighting.”I don’t understand this sentence. It seems as if MI and MEF-SSIM would be inversely correlated to "more informative and suitable for visual perception", which I do not think is the case. The authors have not to be afraid of not being first in all measures. No method, in fusion in general, is better than all the other ones in all measures. Thus I think the authors can simply drop this sentence.

Response:

Thank you very much for your understanding and comments. We deleted this sentence.

Reviewer 2 Report

The authors have addressed most of my concerns.  However, The proposed method uses semantic information, and the related semantic-guided fusion methods should be investigated.

Author Response

Question1:

The proposed method uses semantic information, and the related semantic-guided fusion methods should be investigated.

Response:

Thanks for your suggestion. We added introductions of these three methods in the related work.

Round 3

Reviewer 1 Report

The authors improved the paper over the previous version. They did not include a table with the different quantities meaning as I suggested but anyway the paper has been improved and it's OK with me to accept it after authors answer the following two comments:

Authors say in Section 1.1: “From the perspective of information distribution…”? What do they mean by information distribution?

Authors say about Formula 1 that it is the sum of two cross-entropy losses (BTW “loss” singular is used instead of plural “losses”), but cross-entropy includes the minus sign so the final result is always positive, compare with formula 4. If you want to keep it as it is (a negative quantity, that it makes sense to be maximized, while as cross entropy is always positive you look for its minimum) you should refer it as minus cross entropy.

Author Response

1, Authors say in Section 1.1: “From the perspective of information distribution…”? What do they mean by information distribution?

Response:

Thank you very much for your patience and professionalism. Information distribution indicates the data distribution of the generative model, we have improved the description:

From the perspective of data distribution in image generation, GAN is an implicit generative model and the training data distribution is contained in the trained generator G.

 2, Authors say about Formula 1 that it is the sum of two cross-entropy losses (BTW “loss” singular is used instead of plural “losses”), but cross-entropy includes the minus sign so the final result is always positive, compare with formula 4. If you want to keep it as it is (a negative quantity, that it makes sense to be maximized, while as cross entropy is always positive you look for its minimum) you should refer it as minus cross entropy.

Response:

Sorry for the misinterpretation, we have corrected the mistake and referred it as minus cross entropy.